# Alterations of Selected Hemorheological and Metabolic Parameters Induced by Physical Activity in Untrained Men and Sportsmen

**DOI:** 10.3390/metabo11120870

**Published:** 2021-12-14

**Authors:** Sandor Szanto, Tobias Mody, Zsuzsanna Gyurcsik, Laszlo Balint Babjak, Viktoria Somogyi, Barbara Barath, Adam Varga, Adam Attila Matrai, Norbert Nemeth

**Affiliations:** 1Department of Sports Medicine, Faculty of Medicine, University of Debrecen, Nagyerdei Park 12, H-4032 Debrecen, Hungary; szanto.sandor@med.unideb.hu (S.S.); modytobias@gmail.com (T.M.); gyurcsik.zsuzsanna@med.unideb.hu (Z.G.); 2Doctoral School of Clinical Medicine, University of Debrecen, Nagyerdei krt. 98, H-4032 Debrecen, Hungary; 3Department of Operative Techniques and Surgical Research, Faculty of Medicine, University of Debrecen, Moricz Zsigmond u. 22, H-4002 Debrecen, Hungary; babjakl43@gmail.com (L.B.B.); sogor.viktoria@med.unideb.hu (V.S.); barath.barbara@med.unideb.hu (B.B.); varga.adam@med.unideb.hu (A.V.); matrai.adam@med.unideb.hu (A.A.M.)

**Keywords:** hemorheology, metabolites, physical activity, training, sport

## Abstract

Optimal tissue oxygen supply is essential for proper athletic performance and endurance. It also depends on perfusion, so on hemorheological properties and microcirculation. Regular exercise is beneficial to the rheological status, depending on its type, intensity, and duration. We aimed to investigate macro and microrheological changes due to short, high-intensity exercise in professional athletes (soccer and ice hockey players) and untrained individuals. The exercise was performed on a treadmill ergometer during a spiroergometry examination. Blood samples were taken before and after exercise to analyze lactate concentration, hematological parameters, blood and plasma viscosity, and red blood cell (RBC) deformability and aggregation. Leukocyte, RBC and platelet counts, and blood viscosity increased with exercise, by the largest magnitude in the untrained group. RBC deformability slightly impaired after exercise, but showed better values in ice hockey versus soccer players. RBC aggregation increased with exercise, dominantly in ice hockey players. Lactate increased mostly in soccer players, and the respiratory exchange rate was the lowest in ice hockey players. Overall, short, high-intensity exercise altered macro and microrheological parameters, mostly in the untrained group. Significant differences were found between the two sports. The data can be useful in training status monitoring, selection, and in revealing the causes of physical loading symptoms.

## 1. Introduction

Hemorheological parameters play a pivotal role in tissue perfusion. Blood is a non-Newtonian fluid as its viscosity depends on the shear rate. Lowering the shear rate is associated with increasing viscosity values due to the red blood cell aggregation [1,2,3]. Whole blood viscosity is mainly determined by the plasma viscosity (a Newtonian fluid), number of blood cells, dominantly red blood cells (RBCs), and microrheological parameters of the RBCs, such as deformability and aggregation [1,3,4]. RBC deformability is influenced by the cells’ volume, surface-to-volume ratio, morphology, intracellular viscosity, as well as by cell membrane properties [5,6]. RBC aggregation is determined by plasmatic factors (fibrinogen, other plasma proteins, and macromolecules) and cellular features (deformability, morphology, and the composition of the glyocalyx layer) as well as by the shearing forces [1,4,5,6,7]. It is well known that oxidative stress (free radical reactions), inflammatory processes, mechanical stress, osmolarity changes, oxygenation levels, temperature, nitric oxide, and metabolic and pH alterations may deteriorate these microrheological parameters [5,6,8,9,10,11,12]. Impaired RBC deformability and enhanced RBC aggregation increase viscosity, so decreasing blood fluidity, and result in disturbed tissue perfusion, so increasing vascular resistance, and a deterioration in the microcirculation [3,5,13].

Optimal tissue oxygen supply is essential for proper athletic performance and endurance, an important factor of which is tissue perfusion, and thus, hemorheological parameters and microcirculation characteristics. It is known that exercise, usual physical activity, has a beneficial effect on the rheological status, but it also depends on its type, intensity, and regularity [14,15,16,17].

Romain et al. in their meta-analysis study confirmed that regular exercise decreases hematocrit and RBC aggregation; however, there are still many controversial data and more studies are needed to further analyze these effects [18]. In general, it can be concluded that an unhealthy, sedentary lifestyle with obesity is associated with impaired RBC deformability, enhanced RBC aggregation, and increased hematocrit and plasma viscosity [16,17]. Physical activity is beneficial, but the effect strongly depends on the type of exercise, its intensity, regularity, and duration or volume [16,19,20,21,22]. Irregular, inappropriate, or heavy/intensive physical exercise may increase blood and plasma viscosity, and hematocrit, with bidirectional changes in the microrheological features. Regular, well-balanced exercise may lead to a kind of “hemorheological fitness” characterized with lower blood and plasma viscosity, lower hematocrit, improved RBC deformability, and decreased aggregation [22,23].

The question may rise whether untrained people, who perform physical activity in their leisure time, and professional athletes, sportsmen, may have different hemorheological conditions. What happens with these parameters if short, high-intensity exercises have to be performed? To investigate this question, a standardized condition of physical exercise load can be used, such as the spiroergometry test.

Spiroergometry is a diagnostic procedure to continuously measure respiration and gas metabolism during ergometer exercise [24,25]. The aim of a standard cardiopulmonary exercise testing (CPET) protocol is for the individual to be exposed to a load using an ergometer and incrementally increase workload for about 8 to 12 min until they can go no further. These are often referred to as an incremental ramp protocol to a volitional maximum. It enables judgment of function and performance capacity of the cardiopulmonary system and metabolism. This also makes it possible to determine the maximum exercise capacity, maximal oxygen uptake capacity as well as aerobic threshold and anaerobic/lactate threshold values [24,25].

The aim of our research was to evaluate the macro and microrheological parameters of the blood as well as lactate and respiratory parameters in professional athletes and in untrained men before and after a standardized physical exercise in spiroergometry. We also wished to investigate whether these parameters differ in various professional sports, with different training and competition loads, such as soccer and ice hockey.

## 2. Results

### 2.1. Hematological Parameters

Table 1 summarizes the changes in the general quantitative and qualitative hematological parameters. Short-term, high-intensity exercise by spiroergometry resulted in increased white blood cells (untrained Control: *p* < 0.001, Soccer players: *p* < 0.001, Ice hockey players: *p* < 0.001 vs. before; Control vs. Soccer players: *p* < 0.001; Soccer players vs. Ice hockey players: *p* = 0.01) and platelet counts (Control: *p* < 0.001, Soccer players: *p* < 0.001, Ice hockey players: *p* < 0.001 vs. before) with hemoconcentration (Hct, Control: *p* < 0.001, Soccer players: *p* < 0.001, Ice hockey players: *p* < 0.001 vs. before). The increase in the white blood cell count was the largest in its magnitude in the ice hockey players, while the red blood cell count and hematocrit increased slightly more in the untrained group.

### 2.2. Blood and Plasma Viscosity

Both blood and plasma viscosity increased after the high-intensity exercise. In the case of blood viscosity, the change was significant in all groups (untrained Control: *p* < 0.001, Soccer players: *p* = 0.038, and Ice hockey players: *p* = 0.001). However, the highest elevation was seen in the untrained group. The difference between the untrained groups and the groups of professional sportsmen was more obvious when the blood viscosity values were corrected for 40% hematocrit (Soccer players: *p* = 0.03, Ice hockey players: *p* = 0.034 vs. Control). Hematocrit/viscosity values significantly dropped in the untrained group (*p* = 0.002 vs. before, *p* = 0.004 vs. Soccer players, and *p* = 0.018 vs. ice hockey players) (Figure 1).

### 2.3. Red Blood Cell Deformability

The elongation index at 3 Pa did not change with exercise; however, the values were the highest in ice hockey players (*p* < 0.001 vs. the untrained Control or Soccer players). The maximal elongation index showed a decrease with exercise, more dominantly in the untrained men (*p* = 0.045) and in a smaller manner in the sportsmen groups. The highest shear stress values belonging to the half-maximal elongation index (SS_1/2_ [Pa]) were found in the Soccer player group (*p* < 0.001 vs. untrained Control or Soccer players), associated with the lowest ratio of these two parameters (Figure 2). The ratio of the values tested before and after the exercise are shown in Table 2 with the viscosity data.

Investigating the osmotic gradient deformability (osmoscan) parameters, we found that the maximal elongation index values were the highest in sportsmen (Soccer players: *p* = 0.011, Ice hockey players: *p* < 0.001 vs. Control). With exercise, their values decreased, while the values of ice hockey players increased. The minimal elongation index values were higher in soccer players (*p* = 0.04 vs. Control, *p* = 0.015 vs. Ice hockey players). The parameter derived from the area under the elongation index–osmolarity curves were higher in sportsmen (Soccer players: *p* = 0.023, Ice hockey players: *p* = 0.008 vs. Control), and with high-intensity exercise, these values decreased in soccer players and increased in ice hockey players (Table 3).

### 2.4. Red Blood Cell Aggregation

Table 4 summarizes the various erythrocyte aggregation parameters. Using light-transmission aggregometry M 5 s, M1 5 s, M 10 s, and M1 10 s index parameters were determined. In the untrained group, both M index values (at 0 s^−1^ shear rate) decreased, while M1 index values (at 3 s^−1^ shear rate) increased significantly (M 5 s: *p* = 0.045, M1 5 s: *p* = 0.001, and M1 10 s: *p* < 0.001). Similar changes were observed in the sportsmen groups (Soccer players’ M1 5 s: *p* < 0.001, M1 10 s: *p* < 0.001; Ice hockey players’ M 5 s: *p* = 0.033, M1 5 s: *p* < 0.001, and M1 10 s: *p* = 0.022), however, the largest increase was seen in the untrained control group.

The aggregation values determined by the syllectometry, correlated with the changes of M1 index values, as the aggregation index (AI [%]) increased after the exercise, showing the largest rise in the untrained group (*p* = 0.003). The most stable values were seen in the ice hockey players. Amplitude values were very low in ice hockey players and were the highest in soccer players. After exercise values decreased in all groups, together with the half-time values (t_1/2_ [s]) (*p* < 0.001 in Soccer players).

### 2.5. Maximal Oxygen Uptake, Respiratory Exchange Rate, and Lactate Concentration

The maximal oxygen consumption (VO_2_ max) values were 42.89 ± 4.56 mL/min/kg in the untrained Control group, 58.91 ± 5.67 mL/min/kg in Soccer players (*p* < 0.001 vs. Control group), and 51.66 ± 3.36 mL/min/kg in Ice hockey players (*p* < 0.001 vs. Control group and vs. Soccer players).

The respiratory exchange rate (RER) showed the highest values in the untrained group and was lower in soccer players and the lowest in ice hockey players. The maximal lactate concentration and lactate concentration 5 min after the exercise did not differ significantly; however, when the exercise was finished, the values slightly decreased. The lowest values were seen in the untrained individuals, the highest in soccer players, and in between values were in ice hockey players. The correlation coefficient values between the RER and lactate_max_ were the highest in untrained men and the second highest in ice hockey players (Table 5).

## 3. Discussion

Hemorheological changes induced by exercise is an intensively studied field that is still full of controversies. However, some general conclusions can be taken, thanks to the pioneer groups working on this field in the past two to three decades (e.g., L. Dintenfass, E. Ernst, J.F. Brun, P. Connes, and M. El-Sayed, among others) [16,17,19,21].

Many authors support that regular, low-intensity exercise activity has value for “hemorheological fitness”. As a result of training, the expansion of blood volume, particularly plasma volume, results in better fluidity of the athletes’ blood [15,16,17,21].

Microrheological parameters of red blood cells are influenced by oxidative stress, mechanical stress, metabolic alterations (e.g., accumulation of lactate and decrease in pH), and changes in oxygenation levels [5,6,9,10,11,12,26,27,28]; All are factors present during exercise in various magnitudes. When blood viscosity acutely increases during exercise (as short-term, exercise-induced hyperviscosity), in the case of a healthy vasculature, it means a key modulator for vasodilatation via the endothelial mechanoreceptor-mediated nitric oxide production process, as part of the adaptation [16,17,23,29]. Concerning the effect of lactate concentration increase and RBC deformability impairment [30], it has been demonstrated that red blood cell deformability does not impair under a lactate concentration threshold of 4 mmol/L. Above that, decreased red blood cell deformability can be found [16,31,32]. In athletes with high-level endurance, this correlation is not obvious, probably due to a kind of adaptation to exercise-induced hypoxemia with an improved lactate transfer through the cell membrane [16,31,32,33,34,35,36,37,38]. These findings correlate well with our study. However, the physical fitness level of trained and untrained people can be different, as well as the amount of accumulated lactate that affects lipid peroxidation [31], which influences rheological parameters, such as RBC deformability [5,6]. A wider investigative scope of metabolomics [36] would be important to better reveal the background of these alterations.

In our study, the age and BMI values of the healthy male volunteers were comparable. Therefore, we assumed that the observed changes can be dominantly related to the differences in physical activity level and profession. An important difference between the players of the two sports from the point of view of training physiology is that while soccer players play at medium intensity for a longer period of time, performing a higher proportion of aerobic activity, ice hockey players train at higher intensities at shorter time intervals. This is also supported by the higher VO_2_ max values obtained by spiroergometry in the case of soccer players (better endurance). It is supposed that the differences in the muscle mass and its capillarity and thus metabolic responses might contribute to the explanation of the differences we found [39,40,41]. Hemorheological results (higher EI max values for hockey players under exercise) correlate with the results of maximum lactate levels after exercise and lactate levels measured 5 min after exercise. Presumably, due to the better red blood cell deformability, we may see a better lactate elimination in the case of hockey players as a result of the favorable microcirculation compared to the other two groups. Furthermore, the lower respiratory exchange rate values measured in hockey players can be explained by the positive effect of the high-intensity exercise on lipid and carbohydrate oxidation [33,38].

We have found that professional athletes experience more favorable hemorheological changes as a result of intense exercise than untrained individuals. While hematocrit normalized whole blood and plasma viscosity, and RBC deformability and aggregation deteriorated in the control group, in the case of athletes, these parameters remained unchanged or changed only to a lesser extent.

In our study, hematocrit increased almost equally in all three groups under intense exercise. When viscosity was normalized to 40% hematocrit, a significant increase in whole blood viscosity was observed only in the untreated group, with no increase in athletes relative to preload values. As whole blood viscosity depends primarily on hematocrit, plasma viscosity, and RBC aggregation and deformability [1,2,3], it was hypothesized that a smaller increase in viscosity was more likely to be due to a more favorable change in RBC aggregation and deformability in athletes.

Examining the change in plasma viscosity, a significant increase under load occurred only in the untrained group. An increase was also observed in athletes only with a smaller rate. Alis et al., during high-intensity interval training, observed an increase in serum protein, cholesterol, and triglyceride levels in healthy individuals in addition to an increase in hematocrit, explaining the increase in plasma viscosity [42]. The increase of hematocrit levels in athletes was slightly smaller than in the untrained men in our study, suggesting a lower hemoconcentration, which may explain the more moderate deterioration in plasma viscosity. It should also be emphasized that in our case, the total load time was much shorter than in the above-mentioned study, so no significant fluid loss could develop.

The intense exercise did not change the deformability of RBCs, only in untrained individuals. There are few data in the literature on changes in deformability under short-term intense loading. Kilic-Toprak et al. similarly analyzed hemorheological parameters on female volleyball players before and after the Yo-Yo intermittent recovery test level 1 (Yo-YoIR1) [43]. As a result of the test, an increase in whole blood viscosity and red blood cell deformability was also observed. This contradicts our own results, but may be explained by the different nature of the load, especially the interval type of the load. It is known that the deformability of RBCs improves in healthy adults as a result of regular training. Bizjak et al. also found an increase in the theoretical maximal elongation index at infinite shear stress during six weeks of moderately intense exercise [44].

It was an interesting observation that the value of the RBC aggregation index at rest was slightly higher in hockey players than in soccer players; however, under maximum load, it increased only slightly in hockey players, as opposed to the significant increase observed in soccer players. This difference in the two sports can also be explained by different training and match loads. In hockey, the proportion of short but high intensity loads and the acceleration that requires concentric muscle work is much more common [45].

## 4. Materials and Methods

### 4.1. Volunteers

Thirty-seven male volunteers (ethical permission nr.: DE RKEB/IKEB:5410-2020) took part in the study, forming three different groups. The untrained control group included 11 medical students with a hobby level of physical activity. The second group had 14 professional soccer players (time spent in competitive sport level: 14.4 ± 3.4 years). Twelve professional ice hockey players formed the third group (time spent in competitive sport level: 15.9 ± 4.5 years.). Anthropometric data are presented in Table 6.

### 4.2. Spiroergometry Tests and Collection of Blood Samples

Five minutes before the beginning of the spiroergometry test [25,46,47], we collected venous blood samples from the individuals (median cubital vein, 23 G needle, Vacutainer tubes K_3_-EDTA). After that, all participants had to complete a ramp protocol (Vitamaxima 12) on a treadmill ergometer with increasing performance (W) [46,47]. The protocol started with a 2 min warm-up phase with 4 km/h velocity. In the test phase following the warm-up, the workload of the ergometer increased every minute with 45 W by increasing the speed and elevation of the treadmill. The maximal velocity was 12 km/h reached in the 5th minute and, additionally, the maximal speed gradient was still continuously increasing to a maximum of 17.5 degrees. The reachable maximal workload was 645 W.

During the test, the athletes were connected via face mask to a Vyntus CPX hardver (Vyaire medical, Mettawa, IL, USA), giving minute ventilation, breathing frequency, oxygen uptake, carbon dioxide production, and other specific data. The device has an inbuilt automatic calibration mechanism that we proceeded after every 5th test. The heart rate was measured by a Polar H9 (Polar Electro, Kempele, Finland) chest belt, and real time data were trackable using a Bluetooth connection. The test was performed at a temperature of 20 °C. When participants reached their maximal performance capacity, we stopped the protocol. This running time was 7:04 ± 0:46 min in the control group, 8:19 ± 0:33 min in soccer players, and 6:59 ± 0:43 min in hockey players.

Recorded main parameters were: load [W] and time [min], heart rate [1/min], ventilation volume [L/min], breathing frequency [1/min], oxygen uptake as volume (VO_2_ [mL/min]), carbon dioxide output as volume (VCO_2_, [mL/min]), respiratory exchange ratio (RER, VCO_2_/VO_2_), systolic and diastolic blood pressure [mmHg]. In this paper, the RER values were correlated with the laboratory parameters.

Right after the exercise, we collected venous blood samples again and measured heart rate and blood pressure for 5 additional minutes. In this 5 min resting period, we also collected capillary blood samples from the individuals’ fingertip (right hand, index finger with Accu-Chek Safe T-Pro lancets) to determine the blood lactate concentration in the first and fifth minutes of resting. To test lactate concentration [mmol/L], a Nova Lactate Plus device was used (Nova Biomedical, Waltham, MA, USA).

### 4.3. Laboratory Methods

#### 4.3.1. Hematological Parameters

A Sysmex K-4500 automate (TOA Medical Electronics Co., Ltd., Kobe, Japan) was used to determine red blood cell count (RBC [10^12^/µL]), white blood cell count (WBC [10^9^/µL]), lymphocyte percent (Lymph [%]), granulocyte and monocyte percent (Gr+Mo [%]),]), platelet count (Plt [10^9^/µL]), hematocrit (Hct [%]), hemoglobin concentration (Hgb [g/dL]), mean corpuscular volume (MCV [fL]), mean corpuscular hemoglobin (MCH [pg]), mean corpuscular hemoglobin concentration (MCHC [g/L], and mean platelet volume (MPV [fL]) values.

#### 4.3.2. Hemorheological Parameters

Whole blood and plasma viscosity were determined by a Hevimet-40 capillary viscometer (Hemorex Ltd., Budapest, Hungary) at 90 s^−1^ shear rates. To calculate the whole blood viscosity, the hematocrit count was normalized to 40% according to the Matrai formula [48]: WBV_40%_/PV = (WBV_Hct_/PV)^40%/Hct^ where WBV_Hct_ is the whole blood viscosity [mPas] measured at a 90 s^−1^ shear rate and at the actual hematocrit (Hct [%]); PV is the plasma viscosity [mPas], and Hct is the hematocrit of the sample.

RBC deformability was tested with a LoRRca MaxSis Osmoscan ektacytometer. For the conventional deformability tests, 10 μL of blood was diluted in 2 mL of a polyvinyl-pyrrolidone (PVP)/phosphate-buffered saline (PBS) solution (viscosity: 29.6 mPas, osmolarity: 293, mOsm/kg, and pH: 7.2). The elongation index (EI) values of red blood cells were determined in the function of shear stress (SS [Pa] at a range of 0.3–30 Pa) [49,50]. The individual EI-SS curves were compared using the EI values at 3 Pa, the maximal elongation index (EI_max_) and the shear stress belonging to the half of it (SS_1/2_, [Pa]) and their ratio (EI_max_/SS_1/2_) and calculated with the help of the Lineweaver–Burk equation [51].

RBC osmotic gradient deformability (osmoscan) parameters were determined using 250 μL of a blood sample that was suspended to 5 mL of a PVP–PBS solution. In this measurement, the SS was constant (30 Pa), while the osmolality (O [mOsm/kg]) of the suspending medium changed gradually from 0 to 500 mOsm/kg. The descriptive parameters of the EI-O curves are the followings: the minimal elongation index values measured in a low osmotic environment (EI min), the maximal elongation index values (EI max, not equal to EI_max_ above), the half EI max at high osmolality range (EI hyper), the belonging osmolality vales (O min and O (EI max) and O hyper, [mOsm/kg]), and the area calculated from the individual elongation index-osmolality curves [49,50,52].

RBC aggregation was tested using two devices operating with different methods. A Myrenne MA-1 erythrocyte aggregometer (Myrenne GmbH, Germany) was used to test erythrocyte aggregation by the light transmission method. After disaggregation (shear rate: 600 s^−1^) of a 20 μL blood sample, the aggregation index values were determined at the 5th and the 10th second of the aggregation process in M mode (shear rate: 0 s^−1^) and in M1 mode (shear rate: 3 s^−1^) [49,50]. Accordingly, the parameters were M 5 s, M 10 s, M1 5 s, and M1 10 s index values. In the LoRRca MaxSis Osmoscan ektacytometer (Mechatronics BV, The Netherlands), based on the laser backscattering method, further parameters could be determined. In the Couette system, the blood sample (1 mL) was disaggregated by rotation, and as the rotor stopped the aggregation process started, while a syllectogarm of the reflected laser beam (intensity) was analyzed [49,50]. The obtained parameters were: amplitude (Amp [au]), aggregation index (AI [%]), and half-amplitude time (t_1/2_ [s]).

### 4.4. Statistical Analysis

Data are expressed as means ± standard deviation (S.D.). A SigmaStat for Windows (Systat Software Inc., San Jose, USA) software was used for statistical analysis. The D’Agostino–Pearson normality test was used to determine Gaussian distribution. Intra- and inter-group differences were analyzed by ANOVA test followed by post hoc the Bonferroni test or Dunn’s method, depending on the result of normality test. Before/after relations were also analyzed by paired *t*-test or the Wilcoxon Signed-Rank test, depending on the normality of data distribution. Probability values (*p*) less than 0.05 were considered as statistically significantly different.

## 5. Conclusions

Overall, exercise (physical exercise by spiroergometry) in the untrained group showed more significant metabolic and hemorheological changes. Macro and microrheological differences were found not only in the comparison of the untrained group and the athletes but also in the two sports (ice hockey vs. soccer: shorter and high intensity vs. longer and moderate intensity training and games). Regular, professional sport activity may result in a beneficial hemorheological status that improves tissue perfusion, together leading to a better performance (Figure 3). An examination of the hemorheological parameters can be a useful adjunct in assessing the health status of athletes. The results of our tests may help to monitor individual condition during their team’s regular semiannual diagnostic surveys. In addition, it may provide an opportunity for the coaching staff to modify the training program and training methods, if necessary, taking into account any deviations that can be corrected. However, based on these promising results, a more detailed and more sophisticated metabolomics study will be important.

## Figures and Tables

**Figure 1 metabolites-11-00870-f001:**
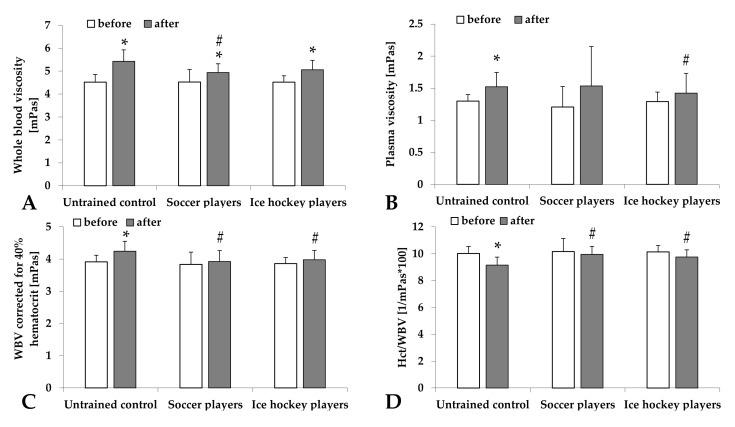
Changes of whole blood viscosity (WBV [mPas]) (**A**), plasma viscosity (PV [mPas]) (**B**), WBV corrected for 40% hematocrit (Hct) (**C**), and Hct/WBV ratio (**D**) in untrained control group and groups of professional soccer players and ice hockey players before and after the standardized physical exercise load by spiroergometry. Means ± S.D.; * *p* < 0.05 vs. before, # *p* < 0.05 vs. untrained.

**Figure 2 metabolites-11-00870-f002:**
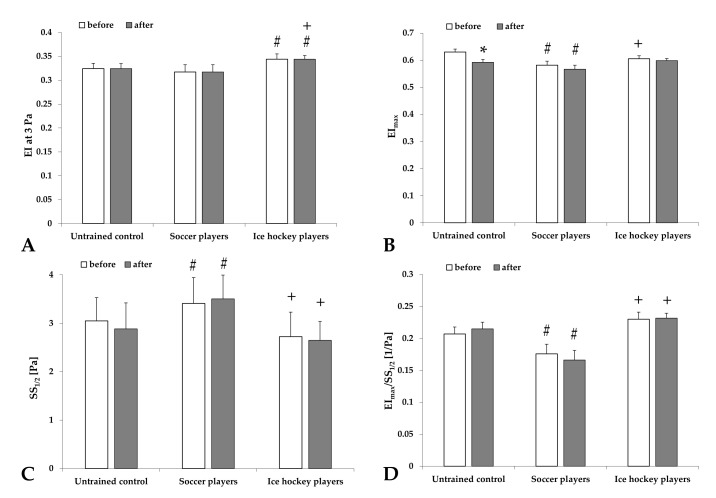
Changes of red blood cell deformability parameters: the elongation index at 3 Pa shear stress (EI at 3 Pa) (**A**), the maximal elongation index (EI_max_) (**B**), the shear stress at half-EI_max_ (SS_1/2_ [Pa]) (**C**), and EI_max_/SS_1/2_ ratio (**D**) in untrained control group and groups of professional soccer players and ice hockey players before and after the standardized physical exercise load by spiroergometry. Means ± S.D.; * *p* < 0.05 vs. before, # *p* < 0.05 vs. untrained group, and + *p* < 0.05 vs. professional soccer players.

**Figure 3 metabolites-11-00870-f003:**
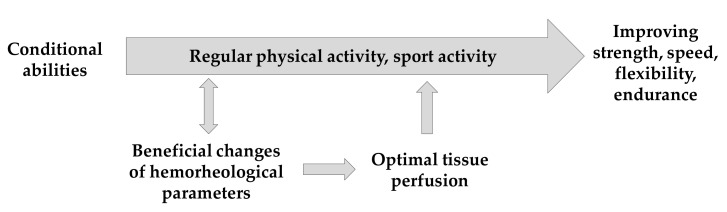
Supposed role in beneficial hemorheological changes supporting regular physical/sport activity leading to better performance.

**Table 1 metabolites-11-00870-t001:** Changes of hematological parameters in untrained control group and groups of professional soccer players and ice hockey players before and after the standardized physical exercise load by spiroergometry.

Variable	UnTrained Control Group			Professional Soccer Players			Professional Ice Hockey Players		
	Before	After	Before/After Ratio	Before	After	Before/After Ratio	Before	After	Before/After Ratio
WBC [×10^9^/L]	6.53 ± 0.99	11.86 ± 2.34 *	1.81 ± 0.22	5.22 ± 1.21 #	9.28 ± 2.39 *,#	1.78 ± 0.37	5.48 ± 0.84 #	11.01 ± 2.08 *,+	2.05 ± 0.52
Lymph [%]	37.36 ± 10.77	46.24 ± 8.16 *	1.28 ± 0.21	32.39 ± 5.52	43.47 ± 8.34 *	1.31 ± 0.21	34.12 ± 7.99 +	35.01 ± 11.77 #,+	1.04 ± 0.27 #,+
Gr+Mo [%]	10.54 ± 2.23	10.21 ± 2.78	0.97 ± 0.16	9.08 ± 2.54 #	8.43 ± 2.43 #	1.06 ± 0.47	13.79 ± 3.85 #,+	11.44 ± 4.45 +	0.82 ± 0.18 #,+
RBC [×10^12^/L]	5.06 ± 0.25	5.45 ± 0.25 *	1.08 ± 0.03	5.24 ± 0.19 #	5.53 ± 0.22 *	1.06 ± 0.03 #	5.14 ± 0.33	5.41 ± 0.41 *	1.05 ± 0.03 #
Hct [%]	45.14 ± 2.11	49.45 ± 2.29 *	1.10 ± 0.03	45.52 ± 1.56	49.02 ± 1.42 *	1.08 ± 0.03 #	45.71 ± 1.74	49.27 ± 2.78 *	1.08 ± 0.03
Hgb [g/L]	15.46 ± 0.84	16.77 ± 0.97 *	1.08 ± 0.03	15.68 ± 0.73	16.53 ± 1.07 *	1.06 ± 0.06 #	15.65 ± 0.68	17.86 ± 6.21 *	1.14 ± 0.41 #
MCV [fL]	89.24 ± 3.05	90.74 ± 2.87 *	1.02 ± 0.01	86.95 ± 3.07 #	88.62 ± 3.06 *,#	1.02 ± 0.01	89.09 ± 3.25 +	91.21 ± 3.90 *,+	1.02 ± 0.01 #,+
MCH [pg]	30.57 ± 1.34	30.76 ± 1.49	1.01 ± 0.01	29.95 ± 1.47	29.91 ± 2.33	1.00 ± 0.06	30.51 ± 1.12	31.98 ± 1.15	1.05 ± 0.21
MCHC [g/L]	34.24 ± 0.72	33.90 ± 0.85	0.99 ± 0.02	34.44 ± 0.89	33.75 ± 2.19	0.98 ± 0.06	34.24 ± 0.54	33.71 ± 0.64 *,+	0.98 ± 0.02
Plt [×10^9^/L]	231.45 ± 49.71	301.59 ± 66.13 *	1.31 ± 0.08	215.46 ± 22.61	287.25 ± 41.43 *	1.31 ± 0.14	224.75 ± 38.89	288.67 ± 45.53 *	1.29 ± 0.10
MPV [fL]	10.77 ± 0.77	11.09 ± 0.91	1.03 ± 0.03	10.35 ± 1.23	10.46 ± 1.19	1.03 ± 0.03	10.40 ± 0.92	10.88 ± 1.00	1.05 ± 0.04

Means ± S.D.; * *p* < 0.05 vs. before, # *p* < 0.05 vs. untrained group, and + *p* < 0.05 vs. professional soccer players.

**Table 2 metabolites-11-00870-t002:** The ratio of blood and plasma viscosity and red blood cell deformability parameters tested before and after the standardized physical exercise load by spiroergometry in untrained control group and groups of professional soccer players and ice hockey players.

Before/After Ratio of	Untrained Control Group	Professional Soccer Players	Professional Ice Hockey Players
WBV [mPas]	1.20 ± 0.08	1.10 ± 0.10 #	1.12 ± 0.05 #
PV [mPas]	1.18 ± 0.19	1.41 ± 0.77	1.09 ± 0.11
Hct_40%_ [%]	1.08 ± 0.06	1.03 ± 0.13	1.03 ± 0.06 #
Hct/WBV [mPas^−1^]	0.91 ± 0.046	0.99 ± 0.11 #	0.96 ± 0.04 #
EI at 3 Pa	1.02 ± 0.11	1.02 ± 0.05	1.02 ± 0.04
EI_max_	1.13 ± 0.55	0.98 ± 0.04 #	0.1 ± 0.04
SS_1/2_ [Pa]	1.02 ± 0.37	0.96 ± 0.14	0.96 ± 0.17
EI_max_/SS_1/2_ [Pa^−1^]	1.34 ± 1.32	1.04 ± 0.11	1.06 ± 0.15

Means ± S.D.; # *p* < 0.05 vs. untrained control group.

**Table 3 metabolites-11-00870-t003:** Changes of osmotic gradient deformability (osmoscan) parameters of the red blood cells in untrained control group and groups of professional soccer players and ice hockey players before and after the standardized physical exercise load by spiroergometry.

Variable	Untrained Control Group			Professional Soccer Players			Professional Ice Hockey Players		
	Before	After	Before/ After Ratio	Before	After	Before/ After Ratio	Before	After	Before/ After Ratio
EI min	0.117 ± 0.007	0.12 ± 0.01	1.029 ± 0.069	0.13 ± 0.013 #	0.128 ± 0.009 #	0.992 ± 0.079	0.122 ± 0.01	0.119 ± 0.008 +	0.979 ± 0.074
EI_max_	0.548 ± 0.011	0.547 ± 0.011	0.999 ± 0.034	0.569 ± 0.015#	0.558 ± 0.007 *,#	0.978 ± 0.022	0.554 ± 0.011 +	0.566 ± 0.008 *,#,+	1.022 ± 0.027 +
EI hyper	0.2742 ± 0.005	0.2738 ± 0.005	0.999 ± 0.033	0.284 ± 0.007 #	0.279 ± 0.004 *#	0.978 ± 0.021	0.277 ± 0.005 +	0.283 ± 0.004 *,#,+	1.021 ± 0.027 +
O min [mOsm/L]	139.09 ± 3.96	145.45 ± 4.69 *	1.04 ± 0.02	137.5 ± 4.381	141.08 ± 4.72 #	1.021 ± 0.03 #	138.66 ± 4.81	140.75 ± 4.22 #	1.015 ± 0.016 #
O (EI_max_) [mOsm/L]	281.54 ± 16.38	278.64 ± 29.33	0.992 ± 0.115	288.64 ± 9.45	291.33 ± 11.63	1.007 ± 0.03	282.5 ± 9.01	289.5 ± 9.67	1.025 ± 0.022
O hyper [mOsm/L]	413.55 ± 14.14	418.82 ± 15.09	1.012 ± 0.016	418.21 ± 15	422.91 ± 16.26	1.008 ± 0.019	424.83 ± 9.97 #	425.5 ± 11.21	1.002 ± 0.015
Area	143.98 ± 5.94	142.97 ± 5.11	0.994 ± 0.034	150.2 ± 6.63 #	147.91 ± 5.91 #	0.98 ± 0.024 #	150.51 ± 4.71 #	153.95 ± 3.54 #,+	1.024 ± 0.038 +

Means ± S.D.; * *p* < 0.05 vs. before, # *p* < 0.05 vs. untrained control group, and + *p* < 0.05 vs. professional soccer players.

**Table 4 metabolites-11-00870-t004:** Red blood cell aggregation parameters in untrained control group and groups of professional soccer players and ice hockey players before and after the standardized physical exercise load by spiroergometry.

Variable	UnTrained Control Group			Professional Soccer Players			Professional Ice Hockey Players		
	Before	After	Before/ After Ratio	Before	After	Before/ After Ratio	Before	After	Before/ After Ratio
M 5 s	3.38 ± 1.13	2.81 ± 1.07 *	0.89 ± 0.36	2.97 ± 1.07	2.65 ± 0.96	0.99 ± 0.64	3.39 ± 0.88	3.00 ± 1.41 *	0.88 ± 0.32
M1 5 s	2.75 ± 1.23	4.07 ± 1.15 *	1.80 ± 1.02	2.78 ± 1.03	3.61 ± 1.01 *,#	1.53 ± 0.94	3.14 ± 1.30	4.02 ± 1.14 *	1.34 ± 0.44
M 10 s	9.43 ± 3.97	9.00 ± 3.08	1.07 ± 0.41	7.62 ± 3.34 #	8.00 ± 2.55	1.27 ± 0.69	8.89 ± 3.36	8.36 ± 3.12	1.03 ± 0.50
M1 10 s	7.27 ± 2.57	10.53 ± 3.41 *	1.61 ± 0.68	6.20 ± 3.07	8.94 ± 2.86 *,#	1.74 ± 0.99	7.98 ± 3.80 +	9.81 ± 3.89 *	1.49 ± 0.97 #
AI [%]	74.83 ± 18.44	86.45 ± 10.12 *	1.24 ± 0.44	63.41 ± 4.90 #	68.31 ± 3.59 *,#	1.09 ± 0.06	78.43 ± 15.65 +	79.19 ± 9.59 #,+	1.03 ± 0.15 #,+
Amp [au]	7.05 ± 4.80	4.55 ± 4.20	0.79 ± 0.29	16.80 ± 2.66 #	15.54 ± 2.26 #	0.96 ± 0.18	1.39 ± 2.95 #,+	0.20 ± 0.27 #,+	0.85 ± 0.90
t_1/2_ [s]	1.91 ± 2.04	0.97 ± 1.01	2.01 ± 3.26	2.25 ± 0.49	1.70 ± 0.29 *,#	0.75 ± 0.15	1.48 ± 1.29 +	1.36 ± 0.68 #,+	1.89 ± 1.96 +

Means ± S.D.; * *p* < 0.05 vs. before, # *p* < 0.05 vs. untrained control group, and + *p* < 0.05 vs. professional soccer players.

**Table 5 metabolites-11-00870-t005:** Respiratory exchange rate (RER) and lactate concentration (maximal and 5 min and their ratio) and their correlation coefficients in untrained control group and groups of professional soccer players and ice hockey players before and after the standardized physical exercise load by spiroergometry.

Group	RER	Lactate_max_ [mmol/L]	Lactate_5′_ [mmol/L]	Lactate_max_/Lactate_5′_	R^2^ of RER and Lactate_max_	R^2^ of RER and Lactate_5′_
Untrained control group	1.22 ± 0.08	12.71 ± 1.91	12.29 ± 2	1.04 ± 0.13	0.4176	0.1319
Professional soccer players	1.18 ± 0.05	14.94 ± 3.01 #	14.58 ± 2.96 #	1.03 ± 0.16	0.0029	0.0287
Professional ice hockey players	1.13 ± 0.04 #,+	13.26 ± 1.96	12.53 ± 2.16	1.07 ± 0.14	0.2027	0.1439

Means ± S.D.; # *p* < 0.05 vs. untrained control group and + *p* < 0.05 vs. professional soccer players.

**Table 6 metabolites-11-00870-t006:** Age, height, weight, body mass index (BMI), and further body composition data (tested with InBody 770 device; InBody USA Co., Ltd., Cerritos, CA, USA) of the participants.

Group	Age [year]	Height [cm]	Weight [kg]	BMI [kg/m^2^]	Percent Body Fat [%]	Skeletal Muscle Mass [kg]
Untrained control group	25.09 ± 2.55	184.09 ± 5.82	89.55 ± 13.02	26.36 ± 3.41	20.41 ± 3.91	38.87 ± 1.94
Professional soccer players	22.71 ± 3.43	182.21 ± 5.63	77.79 ± 6.55 #	23.64 ± 1.08	10.03 ± 4.55 #	39.69 ± 3.54
Professional ice hockey players	24.25 ± 4.29	183.92 ± 5.6	86.17 ± 8.72 +	25.42 ± 1.68 +	16.46 ± 3.94 #,+	43.35 ± 8.83 #

Means ± S.D., # *p* < 0.05 vs. untrained control group and + *p* < 0.05 vs. professional soccer players.

## Data Availability

Because of the participant consent obtained as part of the recruitment process, it is not possible to make these data publicly available. The data presented in this study are available on request from the corresponding author.

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
