# Peer review of "Alterations of Selected Hemorheological and Metabolic Parameters Induced by Physical Activity in Untrained Men and Sportsmen"

_metabolites, 2021, doi:10.3390/metabo11120870_

Round 1

Reviewer 1 Report

This original article could be very interesting according to hemorheological properties and give important information between different subjects. However, the contents seem to be speculative in many points and detailed information is missing in the section on methods Furthermore, authors need to review sufficient literature and should give more information in detail for readers before the publication.   

Abstract

Line 19: Please change to “hemorheological properties and microcirculation”

Line 20: or instead of regulatory “……intensity and volume or duration”?!

At the end of the abstract: I understood that the two different sport types showed macro and micro rheological differences. However, what provided these results regarding the practice in athletes? Please add this crucial point at the end of the abstract.

Introduction

Line 41-42: RBC deformability is also affected by RBC nitric oxide. Please add this point.

Line 49-50: Please write an equal word! “red blood cell or erythrocyte” for the entire manuscript.

Line 57: Please add or change to “volume” and there are many previous studies in which the correlations between VO2max and hemorheological properties. Please describe additionally this point.

Line 58-59: is this acute effect during regular exercise?

Line 64: also, duration or volume.

Line 74-83: references are missing!

Line 82: aerobe, anaerobe please change to “aerobic threshold! and anaerobic/lactate…..”

Results

Generally, you have to increase the statistical power. I would like to see the effect size of your statistical analyses. Please add effect size and confidence interval 95% for all significances. Also, describe these in the section on methods (statistical analyses)

Line 95: Do you mean here p=0.010 ??? A Statistical description such as p=0.001 doesn’t exist. You have to describe and change to p<0.001 (line 106, 155).

Discussion

Line 195 - 201: Additionally, previous study outcomes showed that the aspect between the accumulated lactate concentration and hemorheological parameters is controversial because of the physical fitness level of different subjects e. g. in trained and non-trained persons how accumulated lactate affects lipid peroxidation, which influences rheological parameters e.g. RBC deformability. Please add this aspect.

Line 208 – 211: Did you measure VO2max in your study? where are these values? or do you mean here outcomes of previous studies? VO2max or O2 values are missing.

Line 216 – 217: I would like to suggest that you calculate and add fat and carbohydrate oxidation using the stoichiometry method because you already measured O2 uptake right? Otherwise, this argument is speculative!

Line 230: reference is missing.

Materials and methods

Generally, 4.2. Spiroergometry and collection of blood samples should be described more in detail.

Line 265: How was the gas analyzer calibrated before the test procedure? Please describe it in detail. And why did you choose a Bruce protocol instead of a ramp protocol?

Line 271: Please add the appropriate reference.

Line 284-287: Which analyzer was used to determine the blood lactate concentration and how many venous and capillary blood samples were collected?

Line 333 – 342: The statistical analysis must be made again. Why didn’t you analyze using a one-way ANOVA test (parametric) or Kruskal-Wallis rank test (non-parametric) with appropriate post-hoc test? There are three groups in your study and these 3 groups were compared right?! Also, please add effect size and 95% confidence interval.

Conclusion

Generally, your conclusion is very speculative. What do you mean by professional sports activity? Do you mean sport-specific such as ice hockey? Which professional sports activity? e.g. intensity, duration, and frequency? According to your results, what can you transfer into the practical field for athletes and non-trained persons? Please in detail and carefully!

Author Response

Dear Reviewer,

thank you very much for the review with the very valuable comments. Agreeing with the comments, we have prepared the revision, accordingly. Please also find our detailed answers in the attached file. We hope that the answers and corrections in the revised version might be acceptable.

Sincerely yours,
Norbert Nemeth

Reviewer 2 Report

Szanto and colleagues describe alterations in hemorheological and metabolic parameters in recreational and trained athletes as a function of physical activity (treadmill). Unfortunately, they only measured lactate levels as a proxy for metabolism, while a more comprehensive metabolomics analysis (especially on the samples from professional athletes) would have been really interesting. Some of the papers the authors reference have indeed combined both rheological measurements and metabolomics analyses with interesting insights. I would recommend removing the term “metabolic” from the title, because it is suggestive of data within the manuscript that the authors do not actually show within the manuscript. Overall, it is an interesting study with potentially interesting findings. However, since the paper is submitted for consideration of publication in a thematic journal named “Metabolites”, the manuscript may be better suited for a sports journal than a biochemistry one.

As a metabolomics expert, I felt kind of betrayed myself after noticing the total lack of data on metabolites (apart from lactate) upon acceptance of this review. As such, my bias is that the paper may be accepted as is without the term metabolites in the title in a more sports-oriented journal, or actual metabolic analyses should be performed. Too bad I cannot reveal my identity to the authors as a reviewer, because I would love to run these very samples myself at no cost to the authors in the form of a collaboration.

  • Elevation in blood viscosity after exercise was noted. Can this be also explained by loss of water (e.g., through sweating) during exercise?
  • I would warmly invite the authors to consider maximising the value of the unique sample set they collected by running actual metabolomics analyses on these samples - or at least referencing studies from groups they know well (e.g., Nemkov et al Int J Mol Sci 2021) where rheological measurements and metabolomics were combined. Again, I agree that the cohort enrolled here is unique and the manuscript could be so much more impactful with these data...

Author Response

(The authors gave the same response as above.)

Reviewer 3 Report

Thank you for the opportunity to review your interesting manuscript. Please consider the following comments and suggestions:

  • Thank you for providing details of your ethical approval. I think it would be helpful to describe in detail any inducements which were given to participants, and processes followed to ensure that participation was entirely voluntary. I note that some participants were medical students. Were any of these students taught/supervised/assessed by any of the investigators? If so, the previous comment is particularly important.
  • Please include some quantitative results in the abstract
  • You have described the characteristics of your population in a paragraph, it may be beneficial to convert this to a table, including more detail. As BMI can be misleading in muscular individuals, it would be useful to have weight and height separately (and any other variables you have, such as waist circumference. 
  • Your strategy of using many t-tests would appear to be a form of 'multiple testing', and therefore prone to Type 1 error (especially with a fairly liberal threshold of p<0.05). Please describe approaches you have taken to overcome this problem.

Author Response

(The authors gave the same response as above.)

Round 2

Reviewer 2 Report

Dear Authors, thank you for the polite replies. 
I apologize if my comments came across as aggressive. My offer  ToruÅ„ Metabolomics on this set still stands and I am glad to sign my review, while recommending the publication of the revised manuscript. Congratulations for an interesting study.